# Water Column Optical Properties of Pacific Coral Reefs Across Geomorphic Zones and in Comparison to Offshore Waters

**Brandon J. Russell [1,2,\*], Heidi M. Dierssen [1,\*]**  **and Eric J. Hochberg [3]**

1   Department of Marine Sciences, University of Connecticut, Groton, CT 06340, USA
2   Currently: Labsphere, Inc., North Sutton, NH 03260, USA
3   Bermuda Institute of Ocean Sciences, St. George's GE 01, Bermuda
\*   Correspondence: branjrussell@gmail.com (B.J.R.); heidi.dierssen@uconn.edu (H.M.D.);
    Tel.: +1-203-241-7253 (B.J.R.); +1-860-405-9239 (H.M.D.)

**Abstract:** Despite the traditional view of coral reefs occurring in oligotrophic tropical conditions, water optical properties over coral reefs differ substantially from nearby clear oceanic waters. Through an extensive set of optical measurements across the tropical Pacific, our results suggest that coral reefs themselves exert a high degree of influence over water column optics, primarily through release of colored dissolved organic matter (CDOM). The relative contributions of phytoplankton, non-algal particles, and CDOM were estimated from measurements of absorption and scattering across different geomorphic shallow-water reef zones (<10 m) in Hawaii, the Great Barrier Reef, Guam, and Palau (n = 172). Absorption was dominated at the majority of stations by CDOM, with mixtures of phytoplankton and CDOM more prevalent at the protected back reef and lagoon zones. Absorption could be dominated by sediments and phytoplankton at fringing reefs and terrestrially impacted sites where particulate backscattering was significantly higher than in the other zones. Scattering at three angles in the backward direction followed recent measurements of the particulate phase function. Optical properties derived from satellite imagery indicate that offshore waters are consistently lower in absorption and backscattering than reef waters. Therefore, the use of satellite-derived offshore parameters in modeling reef optics could lead to significant underestimation of absorption and scattering, and overestimation of benthic light availability. If local measurements are not available, average optical properties based on the general reef zone could provide a more accurate means of assessing light conditions on coral reefs than using offshore water as a proxy.

**Keywords:** Inherent Optical Properties (IOPs); coral reef; colored dissolved organic matter (CDOM); absorption; scattering; light ecology; remote sensing

## 1. Introduction

While there is a view that tropical coral reefs inhabit clear, oligotrophic waters [1–5], these systems can exhibit a range of water clarities, from relatively clear oceanic-type to high-attenuation regimes with complex optical properties [6–8]. As coastal environments, reefs are influenced by the input of dissolved organics, particulates, and nutrients from terrestrial systems, as well as those generated in place and advected from offshore. Some important research has been carried out describing light environments and attenuation on selected coral reefs [1,9–14] or over small regional scales [15]. This study builds upon that research to present a snapshot of water column optical properties measured across diverse reef systems of the Pacific Ocean to better understand large-scale influences on water clarity and light regimes influencing the benthos.

The intensity and spectral quality of light available at the benthos is determined by the absorption and scattering of photons as they travel downwards through the water column. These processes are defined by the Inherent Optical Properties (IOPs) of the water, which vary according to the composition, concentration, and size of suspended particles and dissolved material [16]. The magnitude of measured absorption $a(\lambda)$, scattering $b(\lambda)$, and beam attenuation $c(\lambda)$ [= $a(\lambda) + b(\lambda)$] coefficients, as well as their spectral shapes and relative ratios (Table 1) can be related to specific biogeochemical and compositional properties of the optically active constituents [16–22]. Once measurements are corrected for the effect of water itself, each IOP may in theory be partitioned linearly into estimates of dissolved, phytoplankton, and non-algal particulate fractions [22], which are the main influences on water column optics.

**Table 1.** Summary of measured and derived optical data for all stations in this study.

| Parameter | Unit | Definition | Method |
|---|---|---|---|
| $a(\lambda)$ | m$^{-1}$ | Total absorption coefficient (=400–750 nm) [1] | unfiltered ac-s |
| $a_{pg}(\lambda)$ | m$^{-1}$ | Non-water absorption coefficient | unfiltered ac-s, corrected |
| $a_g(\lambda)$ | m$^{-1}$ | Colored Dissolved Organic Matter (CDOM) absorption coefficient | filtered ac-s |
| $a_p(\lambda)$ | m$^{-1}$ | Particulate absorption coefficient | filtered and unfiltered ac-s |
| $a_{ph}(\lambda)$ | m$^{-1}$ | Phytoplankton absorption coefficient | derived |
| $a_{nap}(\lambda)$ | m$^{-1}$ | Non-algal particulate absorption coefficient | derived |
| $c(\lambda)$ | m$^{-1}$ | Total attenuation coefficient | unfiltered ac-s |
| $c_{pg}(\lambda)$ | m$^{-1}$ | Non-water attenuation coefficient | unfiltered ac-s, corrected |
| $c_g(\lambda)$ | m$^{-1}$ | Dissolved attenuation coefficient | filtered ac-s |
| $c_p(\lambda)$ | m$^{-1}$ | Particulate attenuation coefficient | filtered and unfiltered ac-s |
| $b_p(\lambda)$ | m$^{-1}$ | Particulate scattering coefficient | unfiltered ac-s |
| $b_{bp}(\lambda)$ | m$^{-1}$ | Particulate backscattering coefficient (=470, 532, 660 nm) | ECO VSF-3 |
| $b_{bp}/b_p(\lambda)$ | | Particulate backscattering ratio (=470, 532, 660 nm) | ECO VSF-3 and ac-s |
| $\beta(\theta,\lambda)$ | m$^{-1}$ | Volume scattering function ($\theta$ = 104°, 130°, 151°; =470, 532, 660 nm) | ECOVSF-3 |
| $S_g$ | nm$^{-1}$ | Slope of dissolved absorption | filtered ac-s |
| $Y$ | | Slope of particulate backscattering | ECO VSF-3 |
| $\gamma$ | | Slope of particulate attenuation | unfiltered ac-s |
| [Chl $a$] | mg m$^{-3}$ | Estimated chlorophyll $a$ concentration | derived $a_{ph}(\lambda)$ |
| $n_p$ | | Particulate index of refraction | Derived |

[1] This wavelength range applies to all of the ac-s measurements.

Colored dissolved organic matter strongly absorbs light, particularly in the UV (290–400 nm) and blue spectral regions. Sources of CDOM include exudate from primary producers such as coral xoozanthellae, algae, seagrasses, and phytoplankton [3,23–28], as well as generation by microbial activity [29,30]. A major source of dissolved organic nitrogen (and related CDOM) is $N_2$ fixation by cyanobacteria, which can be abundant on reef flats and lagoonal patch reefs [31,32]. However, the amount of dissolved organic matter released is highly variable among producers [33–36]. Further, CDOM can be delivered from exogenous sources such as rivers, runoff, and groundwater [16,37,38], which may also deliver highly colored particulate sediments and dissolved nutrients.

Absorption of photosynthetically active radiation (PAR) by phytoplankton will reduce the amount and spectral quality of light reaching the benthos [16,39,40]. In addition to the reduction of PAR, high levels of phytoplankton can impose other biogeochemical stresses on coral reef systems [41,42]. However, past research suggests that suspended chlorophyll levels are typically low around reefs. For example, a recent compilation of data over the Great Barrier Reef showed that chlorophyll $a$ levels were generally low (chl $a \leq 1$ mg m$^{-3}$) for the period 1992–2009 [43].

Non-algal particles absorb and scatter light according to their composition, which is highly variable [16]. On reef systems, the calcium carbonate skeletons of coral, as well as the structures of coralline algae, are broken down to fine particulate sand by physical and biological weathering [44–46]. Suspension of sediments through wind, wave, or other mechanisms increase the water column particulate load, limiting benthic PAR [14,47,48] in addition to other negative impacts on corals [49,50]. Particulate organic matter, i.e., detritus or marine snow, may remain suspended for long periods of time and can be produced on reefs from a variety of sources including coral mucus [51]. Terrestrially sourced particulates such as river input or run-off can increase particle load, limiting PAR and smothering or physically burying corals [37,38].

In addition to defining the light regimes under which coral reefs exist, the optical properties of the water column must be accounted for in remote sensing studies of coral reefs and other benthic constituents [8,52–56]. Many remote sensing studies presume that reef waters are clear and that the optical properties of oligotrophic offshore waters can be used as a proxy for reef systems [57]. For example, the Lyzenga [58] method and similar regression methods for obtaining bathymetry [59] presumes that the optical properties over deep water are the same as those over the reef system. However, as described above, limited evidence suggests that the optical properties of reef waters can be impacted by the release of dissolved material from the coral reef itself. Here, we further evaluate these assumptions by measuring IOPs across a variety of reef systems in Hawaii, Guam, Palau, and portions of the Great Barrier Reef. We compare the optical properties measured across regions, as well as geophysical environments based on reef zonation, to determine the main drivers of water clarity and make comparisons to offshore waters.

## 2. Materials and Methods

### 2.1. Data Collection

The data was collected as part of validation efforts for the COral Reef Airborne Laboratory (CORAL) campaign (more information available at coral.jpl.nasa.gov). In situ data collection occurred between June 2016 and May 2017 (Table 2) during 6 discrete field campaigns across sites in Kaneohe Bay Hawaii (2 different campaigns), Guam, Palau, and Lizard and Heron Islands located along the Great Barrier Reef of Australia (Figure 1). Collection methodology was similar for all campaigns and data presented here are publicly available on SeaBASS [60].

**Table 2.** Locations and dates of the field campaigns and the number of high quality stations sampled.

| Campaign | SeaBASS ID | Dates (Local) | Location | Stations |
|---|---|---|---|---|
| Hawaii Summer 2016 | CHI1606 | 11–19 June 2016 | Kaneohe Bay, Oahu, Hawaii, USA | 32 |
| Lizard Island | CAL1609 | 4–10 September 2016 | Great Barrier Reef, Queensland, Australia | 26 |
| Heron Island | CAH1609 | 17–22 September 2016 | Great Barrier Reef, Queensland, Australia | 27 |
| Hawaii Winter 2017 | CHI1702 | 15–24 February 2017 | Kaneohe Bay, Oahu, Hawaii, USA | 34 |
| Guam | CGU1704 | 9–16 April 2017 | Guam, USA | 27 |
| Palau | CPA1705 | 2–13 May 2017 | Koror, Palau | 26 |

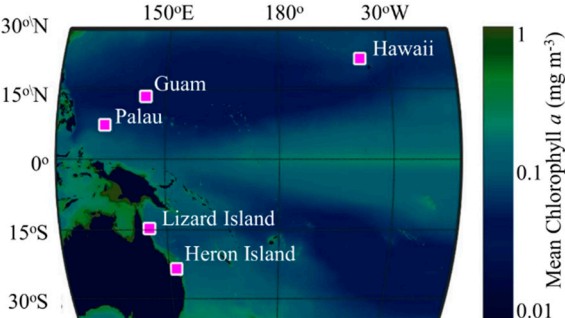

**Figure 1.** Location of COral Reef Airborne Laboratory (CORAL) in-situ validation efforts overlain on a MODIS mission climatology of Chlorophyll *a*.

The primary purpose of in-situ collection was validation of the CORAL airborne data and derived products. In-water optical stations were selected to capture, as fully as possible, the range of benthic and optical environments within each study region (Figure 2).

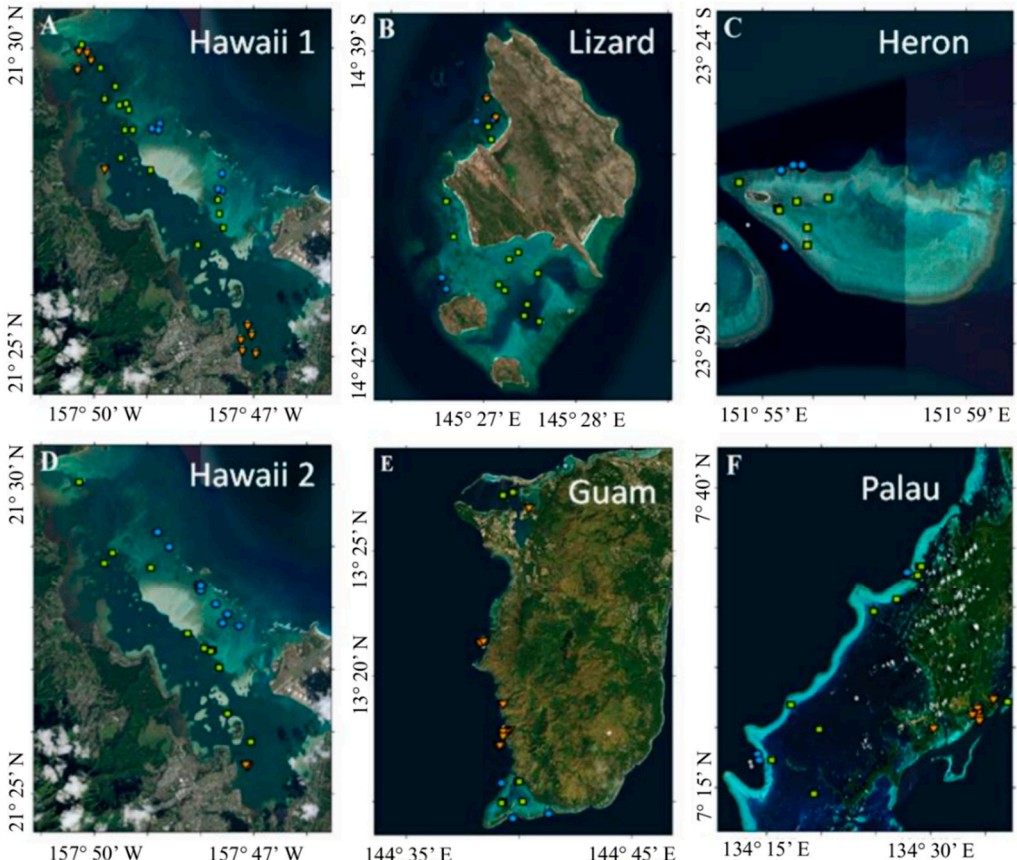

**Figure 2.** Optical measurement stations during Kaneohe Bay during 2016 (**A**), Lizard Island (**B**), Heron Island (**C**), Kaneohe Bay during 2017 (**D**), Guam (**E**), and Palau (**F**) campaigns. Markers correspond to one sample (●), 2–4 samples (□), or 5–7 samples (#). Colors correspond to geomorphic reef zones (see below). Larger images are available in the Supplementary Information (Figures S1–S6).

Broadly, stations included fore, back, fringing, and patch reef zones, as well as lagoons, river mouths, and nearshore coastal areas (Figures S1–S6). Benthic environments included varying mixtures of live and dead coral, coralline rubble and pavement, multiple types of benthic micro- and macro-algae, seagrasses, and carbonate and basaltic sediments.

A WETLabs (Philomath, OR) optical instrumentation package consisted of an ac-s spectrophotometer (25 cm pathlength, approximately 85 wavelengths between 400–750 nm depending on instrument and factory calibration), a VSF-3 scattering meter (wavelengths: 470 nm, 532 nm, 660 nm; angles 104°, 130°, 151°), DH4 data logger, a Seabird (Bellevue, WA) SBE37 CTD and SBE16 pump (deployed on the ac-s), and battery pack. For each regional campaign, daily cruises were made in small boats (<4 m) for data collection. At the end of each cruise, the instrument package was returned to the lab for cleaning, data download, general maintenance, and overnight storage.

At each sampling location, two casts were carried out sequentially: unfiltered and filtered. During the unfiltered cast, coarse mesh screens (0.8 mm) were attached to the ac-s inlets to exclude very large particles. For the filtered cast, 0.2 µm filters (PALL, Port Washington, NY, USA) were attached for measurement of the dissolved fraction. The depth of casts varied based on individual station parameters. The majority of sampling locations (65%) were shallow (<5 m) reefs subjectively judged to have well-mixed water columns and heterogeneous bottom features including calcium carbonate sand flats, seagrasses, and coral boulders and outcroppings. Consequently, optical measurements were typically made at approximately 1–2 m, depending on depth, wave conditions, and bottom rugosity. The actual depth of measurement varied considerably during many casts due to wave action, both by variation in sea-surface height and pitching of the boat itself. Cast durations varied but were

at least 3 min. At locations where stratification or significant depth-dependent variation in optical properties was expected, the instrumentation package was vertically profiled to within ~1.5 m of the bottom where possible. Analysis of these profiles, as well as potential ramifications for reef bio-optics, is ongoing.

Ancillary data including water depth, wind speed, cloud cover, wave height, and (subjective) bottom type were collected for each location. Optical and ancillary data, as well as associated instrument calibration files, are available freely [60] from NASA (SeaBASS).

## 2.2. Data Processing

Handling of IOP data followed the established methods of [61] and [62]. Raw binary data from ac-s, VSF-3, and CTD were integrated using the DH4, merged by time stamp, and processed using WETLabs Archival Processing software. For depth profiles, data were merged based on CTD depth. Manufacturer-supplied temperature correction and clean water calibration coefficients were applied at this stage to ac-s data, with scattering weighting functions applied to the VSF-3.

For the ac-s, temperature and salinity corrections were applied based on CTD data. As some degree of daily variation and instrument drift was observed, a temperature-corrected pure water blank (Milli-Q ultrapure system or equivalent), taken approximately daily, was used to compensate for instrumentation variability [61] and pure water absorption. A residual scattering correction [63,64] was applied to absorption data. The unfiltered cast was used to derive $a_{pg}$ and $c_{pg}$, with data from the filtered cast used for $a_g$. All spectra were interpolated to 1 nm and smoothed using a moving mean with a 5 nm window over the region 400–700 nm. Outlying spectra (>3 standard deviations), if present, were likely the result of bubbles or small particles entrained in the ac-s tubes and were removed. A minimum-value baseline correction was applied to spectra with negative values.

For the VSF-3, seawater scattering and pathlength absorption corrections were applied, and measured dark count values were subtracted from each channel [62]. Outlying scatter data were removed from the data and median $\beta(\theta,\lambda)$ values were obtained at 104°, 130°, and 151° and 470, 532, and 660 nm. Backscattering for each wavelength, $b_b(\lambda)$, was calculated by multiplying by $2\pi\sin\theta$ to convert to polar steradian area, fitting a 3rd order polynomial to the three angles plus a 4th value at 180° ($\sin(\pi) = 0$), and integrating from 90–180° using trapezoidal integration [21]. The phase function was calculated by normalizing the $\beta(\theta,\lambda)$ to the backscattering values. $\beta(151,660)$ was found to be too low compared to the phase functions estimated for the blue and green wavelengths (Figure S7) and there was >20% disagreement between the calibration factors for this angle and wavelength. Hence, the uncertainty in the red backscattering data was considered too high to be included in estimates of Y.

## 2.3. Data Screening

The remote locations investigated by this project necessitated the use of small (~3 m long) boats and relatively large instrument packages deployed over shallow, wave-influenced environments. Due to the shallow depth of many sites, the commonly utilized method of forcing trapped air out of the ac-s by deploying to several meters before data collection could not be employed. At such locations, bubbles in the instrumentation were ameliorated by careful equipment handling. Moreover, frequently cresting waves were observed to inject bubbles into the paths of the instruments for some of the stations. Mechanical shock associated with wave action also introduced noise in optical instrumentation. Both ac-s and VSF were occasionally in very close proximity to the benthos, which could produce man-made plumes of resuspended particles. Hence, data were collected over stations that were suboptimal and a careful screening process was conducted to ensure that only stations with the highest quality data were included in this study. Of the 246 stations, 74 stations were excluded because they had one or more questionable parameters and the remaining 172 stations were considered of high quality and used for this analysis.

### 2.4. Inherent Optical Properties and Derived Parameters

Optical data measured at each station (Table 1) include the non-water absorption coefficient ($a_{pg} = a_t - a_w$) from the unfiltered cast, the dissolved absorption coefficient ($a_g$) from the filtered cast, the non-water attenuation coefficient ($c_{pg}$) from the unfiltered cast, the particulate absorption ($a_p = a_{pg} - a_g$), the scattering coefficient ($b_p = c_p - a_p$) and volume scattering function ($\beta$) at 3 angles and 3 wavelengths. Estimates of the spectral absorption coefficients of phytoplankton ($a_{ph}$) and non-algal particles ($a_{nap}$) were derived from particulate absorption ($a_p$) following the methods of [18] and [65]. The method separates the components of absorption by assuming a ratio of phytoplankton absorption at two discrete wavelengths (412 and 440 nm), and modeling an exponential curve for non-algal particulates based on residual absorption at 412 nm. The decomposition presumed a spectral slope of non-algal particulate absorption of 0.011 and the ratio of $a_{ph}$ at 412nm to 440nm was 0.9. An example of this decomposition for a representative station shows the measured variables as solid lines and the derived as dotted lines (Figure 3).

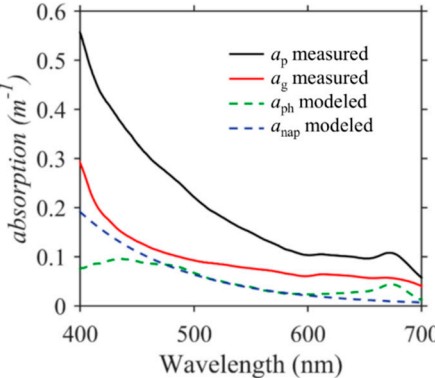

**Figure 3.** Example station showing the measured spectral particulate and dissolved absorption coefficients ($a_p$ and $a_g$) and the modeled results from partitioning $a_p$ into absorption by phytoplankton ($a_{ph}$) and non-algal particles ($a_{nap}$).

Various slope parameters were calculated from the spectral IOP data. The slope of absorption by dissolved organic matter (*S*) was calculated using two approaches. The first, most commonly applied method, involves fitting a single exponent model [22,61]

$$a_g(\lambda) \ = \ a_g(\lambda_0)e^{S_g(\lambda_0 - \lambda)} \tag{1}$$

from 400 to 650 nm, where the reference wavelength $\lambda_0 = 440$ nm. An alternative, potentially more accurate approach, utilizes a hyperbolic model fitted to $a_g$ [66]:

$$a_g(\lambda) \ = \ A((\lambda)/532))^{-S_h} \tag{2}$$

from 400 to 650 nm, where A is an amplitude proxy for CDOM concentration at 532 nm (Figure S8) and $S_h$ is the hyperbolic fit slope of CDOM absorption. While both methods are utilized here, the analysis and discussion will focus on the Single Exponent Model (SEM) calculation of $S_g$ in order to provide a comparison with CDOM absorption slopes measured historically and for use in semi-analytical inversion modeling.

The slope of particulate backscattering (*Y*) was calculated by fitting:

$$b_{bp}(\lambda) \ = \ b_{bp}(\lambda_0)(\lambda/\lambda_0)^{-Y} \tag{3}$$

where the reference wavelength $\lambda_0 = 532$ nm [67]. Values less than 0 were considered questionable and excluded from the analysis.

The spectral slope ($\gamma$) of particulate attenuation ($c_p$) may be related to the particle size distribution (PSD) [68–70] and was calculated with a hyperbolic fit following Boss et al. (2001) and Diehl and Haardt [71]:

$$c_p(\lambda) = A((\lambda)/660))\lambda^{-\gamma} \tag{4}$$

from 400 to 650 nm, where $A$ is the corresponding amplitude of $c_p$ at the reference wavelength of 660 nm. All slope fitting was applied in a nonlinear, least squares method [66].

The IOP data can be related to biogeochemical parameters using bio-optical algorithms which may be regionally tuned. Here, standardized relationships are used. An estimate of in-situ chlorophyll *a* concentration (mg m$^{-3}$) was made utilizing a power model [72]:

$$[Chl\ a] = \left(a_{ph}(\lambda)/0.0132\right)^{1.0967} \tag{5}$$

where $a_{ph}(\lambda)$ is the line-height-corrected phytoplankton absorption at 676 nm [73].

An estimate of the refractive index of bulk particulates relative to seawater, which provides information about the composition of particulates, was modeled following Twardowski et al. (2001) at 532 nm:

$$\hat{n}_p\left(\tilde{b}_{bp}, \gamma\right) = 1 + \tilde{b}_{bp}^{\ 0.5377+0.4867(\gamma)^2}\left[1.4676 + 2.2950(\gamma)^2 + 2.3113(\gamma)^4\right] \tag{6}$$

where $\hat{n}_p$ is the modeled bulk particulate index of refraction, $\tilde{b}_{bp}$ is the particulate backscattering ratio, $b_{bp}/b_p$, at 532 nm and $\gamma$ is the hyperbolic slope of particulate attenuation (Equation(4)).

Total Suspended Matter concentration ([TSM]) can be estimated by:

$$[TSM] = b_{bp}/b_{bp}{}^* \tag{7}$$

where $b_{bp}$ is the particulate backscattering at a given wavelength and $b_{bp}{}^*$ is a mass-specific constant particulate backscatter coefficient, which is highly variable depending on particulate populations (Nechad et al. 2010), although relatively insensitive to wavelength [74,75]. Following [76], $b_{bp}{}^*$ was set at a mean value of $0.0156 \pm 0.009$ m$^2$ g$^{-1}$, with a reference wavelength of 532 nm.

## 2.5. Offshore Data Retrieval

For comparison to reef conditions, satellite-derived IOPs for adjacent offshore waters were retrieved for each campaign. NASA MODIS monthly average data (4 km pixels) of absorption due to gelbstoff and detritus ($a_{dg}$) and absorption due to phytoplankton ($a_{ph}$) from the GIOP model [77] for the time period of each campaign were downloaded using NASA SeaDAS. These satellite products cannot be accurately retrieved over the optically shallow reef waters, but generally, have a low uncertainty over oligotrophic waters <10% [78,79]. The Region of Interest (ROI) for each campaign varied based on local geography, but the size varied between approximately 12 and 24 km offshore from each campaign location. The average and standard deviation of the GIOP $a_{dg}$ satellite-derived product was compared with the sum of the $a_g$ and $a_{nap}$ measured over the neighboring reef waters.

## 2.6. Statistical Analysis

Data between campaigns and geomorphic zones were compared using box plots, where the tops and bottoms of each box represent the 25th and 75th percentiles of the samples, respectively, and the distances between the tops and bottoms are the interquartile ranges. The line in the middle of each box is the sample median. If the median is not centered in the box, it shows sample skewness. The whisker lines extending above and below each box are drawn from the ends of the interquartile ranges to the furthest observations within the whisker length. Observations more than 1.5 times the interquartile range away from the top or bottom of the box are considered outliers, and extreme outliers are removed from the plots for clarity. Statistical significance tests were conducted using the Wilcoxon rank sum test for equal medians. The two-sided rank sum tests the hypothesis that two independent samples, in

the vectors X and Y, come from distributions with equal medians, and returns the *p*-value from the test. *p*-values < 0.05 were considered significant.

## 3. Results

The mean spectral optical properties from the 172 stations are shown in Figure 4. Minimum, maximum, and median value of optical parameters across geomorphic zones collected during this study (Table S13) and mean collected data by class (File S14) are presented in the Supplementary Materials.

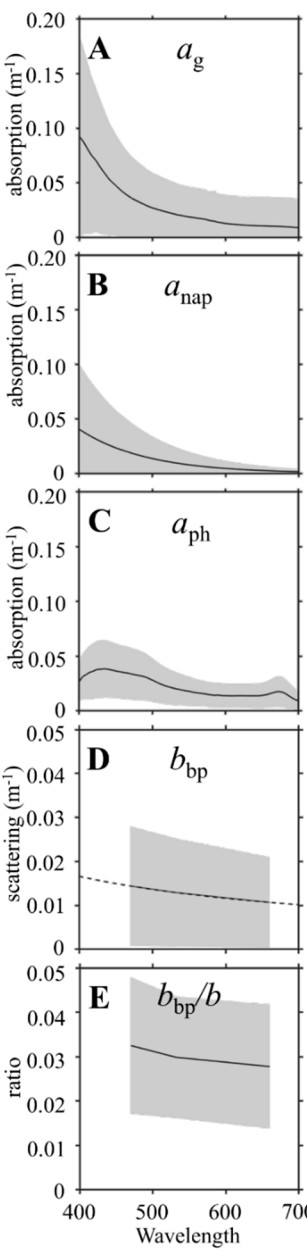

**Figure 4.** Mean spectra measured across all sites for (**A**) absorption by colored dissolved organic matter ($a_g$), (**B**) non-algal particles ($a_{nap}$), and (**C**) phytoplankton ($a_{ph}$), (**D**) particulate backscattering ($b_{bp}$) and (**E**) backscattering ratio ($b_{bp}/b$). Shaded areas are ±1 standard deviation. The dashed line (D) represents the slope of backscattering $Y$.

The variability in the measured and derived optical properties is presented and discussed below in terms of the different regions sampled at a single point in time in comparison to offshore locations

during a similar time frame, and across geomorphic zones. Each sampling location was assigned to one of six geomorphic zones (Table 3) based on prevailing reef classifications [80–82].

**Table 3.** Distribution of high-quality stations collected for the six geomorphic zones and the subsequent grouping into three broader categories.

| Geomorphic Zone | Description | Stations |
|---|---|---|
| Fore Reef | Seaward of main wave breaking zone | 54 |
| Back Reef | Behind reef crest | 26 |
| Shallow Lagoon | Inside reef crests, <10 m depth | 41 |
| Operationally Deep Lagoon | Inside reef crests, >10 m depth | 13 |
| Fringing Reef | Coral directly adjacent to land mass | 17 |
| Terrestrial | Areas where water column or benthic properties are likely to be dominated by terrestrial inputs, such as near river mouths | 21 |

While coral reefs are globally complex and diverse systems, there are strong similarities in geomorphic zonation across Indo-Pacific reefs [83], allowing definition of general classes applicable across all campaigns for examination of large-scale, ecologically relevant differences in optical properties (Figure 5A). Data collected within these six categories are presented in the Supplementary Materials. For the remainder of this analysis, these six zones were broadly grouped into three types of reefs from most oceanic to terrestrially-influenced (Figure 5A):

- Fore reef, which is most exposed to open ocean waters (n = 54)
- Back reef, shallow lagoons and deep lagoons (n = 80)
- Fringing reefs and terrestrial reefs which are impacted by land and river runoff (n = 38).

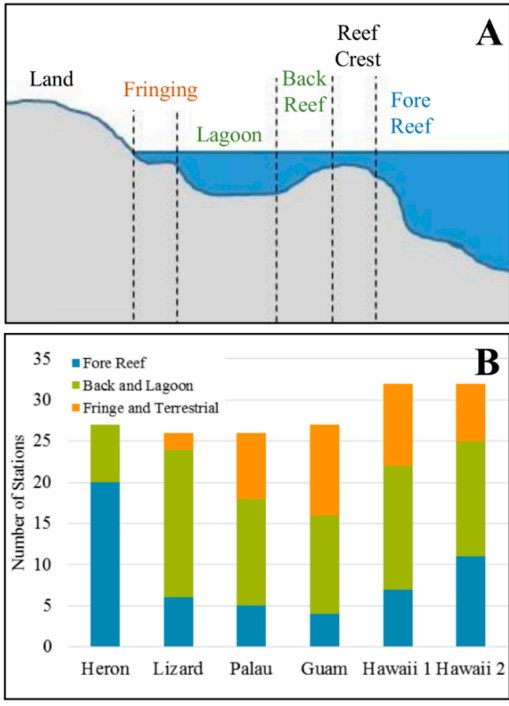

**Figure 5.** (**A**) A visual depiction of the different tropical coral reef zones going from the outer fore reef most influenced by oceanic water to the nearshore fringing and terrestrially impacted zone. (**B**) The number of stations sampled from each campaign with the three broad geomorphic zones color coded similarly to panel (**A**).

With the exception of Heron Island, sampling was conducted regionally within these three broad categories (Figure 5B) and the number of stations within each group is deemed sufficiently diverse across zones to conduct statistical comparisons between the groups.

### 3.1. Interregional Comparison

Considerable variations in absorption and backscattering properties were found between regional campaigns (Figure 6).

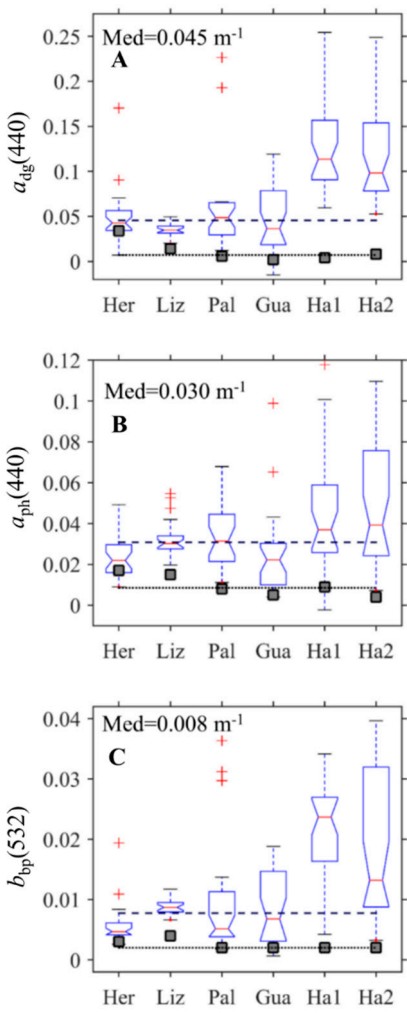

**Figure 6.** Boxplots of the optical properties measured at Heron Island (Her), Lizard Island (Liz), Palau (Pal), Guam (Gu), and Hawaii summer (Ha1) and Hawaii winter (Ha2) in comparison to MODIS imagery collected offshore for the same time period (gray squares) for (**A**) absorption by colored dissolved organic matter and non-algal particles at 440 nm ($a_{dg}$); (**B**) absorption by phytoplankton at 440 nm ($a_{ph}$); and (**C**) particulate backscattering at 532 nm ($b_{bp}$). The median values are shown as lines and in text for all campaigns (Reef) and MODIS values (Offshore). Extreme outliers were removed for clarity.

Median derived dissolved and non-algal absorption ($a_{dg}$) at 440 nm was comparable across Heron, Lizard, Palau, and Guam campaigns with a value of 0.045 m$^{-1}$, but values were higher and more variable for both campaigns in Hawaii, with median values approximately double compared to other sites (>0.1 m$^{-1}$). Differences were less pronounced for absorption by phytoplankton ($a_{ph}$) with a median of 0.030 m$^{-1}$. Hawaii campaigns also showed higher variability in $a_{ph}$ across stations. Particulate backscattering ($b_{bp}$) at 532 nm, an indicator of phytoplankton and sediment resuspension [84] was also

quite variable across sites with a median value of 0.008 m$^{-1}$. Heron Island had the lowest magnitude and variability in $b_{bp}$, while both Hawaii campaigns had the highest amount and variability. The impact of the reef benthic community on water column $b_{bp}$ is particularly evident on the Kaneohe Bay, Hawaii barrier reef, which is known to produce and resuspend fine carbonate sediments [26,85–87].

These data were collected over the span of a year, and biases due to seasonality or individual events (e.g., storms, tidal cycles) may be present between sites or campaigns and will contribute to variability in the IOPs between sites. The Hawaii region (Kaneohe Bay) was the only one to be visited twice, once during summer and once during winter seasons (June 2016 and February 2017). While the sites profiled were not identical, the overall spatial and geomorphic zone distributions were very similar for most of the IOPs (Figure 6) and quite different from the other reefs. As these are snapshots in time, we cannot generalize about the true variability in IOPs in these regions, but we can compare them to offshore waters during a similar time period.

While coral reefs are typically found in "clear," low-latitude waters with high solar irradiance and relatively low nutrient load, reef optical properties are statistically different from oceanic IOPs (Figure 6). Across all sites, the IOPs derived at the reef locations are much higher than those derived over the open ocean from the MODIS satellite. The difference is much higher than the 10% accuracy of such IOP algorithms [78]. Values of $a_{dg}(440)$ in Hawaii during June 2016, for example, were 26 times higher than offshore, while $a_{ph}(440)$ and $b_{bp}(532)$ were 3 and 10 times higher, respectively. Across the entire data set, estimates of absorption and scattering were at least double satellite-derived retrievals of offshore data collected during the same period (Figure 6). This indicates that oceanic waters cannot be used as a proxy for optical properties of reef waters.

### 3.2. Comparison of IOPs across Geomorphic Zone

Median IOPs measured during the entirety of this study reveal potentially unique optical domains across geomorphic zones and illustrate the range of optical conditions on these reefs (Figures S8–S10). Absorption parameters were broadly comparable across fore reefs and back/lagoon geomorphic zones, except for fringing/terrestrially influenced sites (Figure 7A–D).

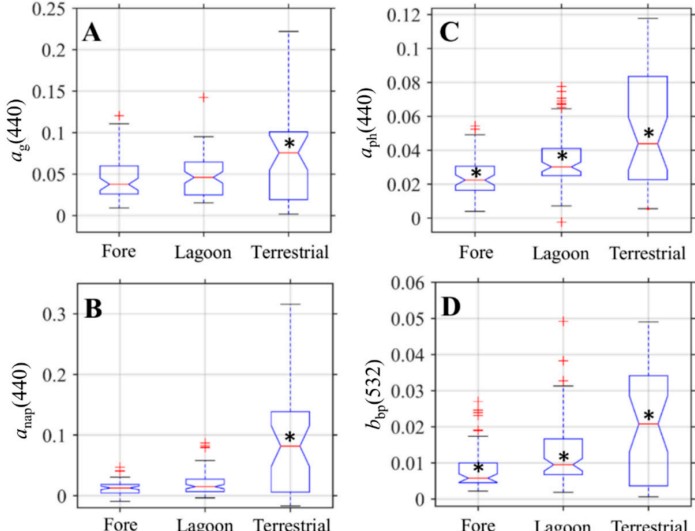

**Figure 7.** Boxplots showing the variability of the water column optical properties aggregated by three broad geomorphic zones: fore reef (Fore), back reef and lagoon (Lagoon), and fringing and terrestrially-impacted reef (Terrestrial). Properties include: (**A**) absorption by colored dissolved organic matter at 440 nm ($a_g$); (**B**) absorption by non-algal particles at 440 nm ($a_{dg}$); (**C**) absorption by phytoplankton at 440 nm ($a_{ph}$); and (**D**) particulate backscattering at 532 nm ($b_{bp}$). Starred values indicate that the values were significantly different ($p < 0.05$) in paired tests with the other two zones.

Dissolved and non-algal particle absorption ($a_g$, $a_{nap}$) at 440 nm was significantly higher at fringing/terrestrial sites compared to other classes. Phytoplankton absorption followed a statistically significant gradient from highest in the fringing/terrestrial where nutrient levels may be highest, to middle values in the back/lagoons where water column phytoplankton may be more protected to the lowest absorption values at the fore reef sites most influenced by low-nutrient oceanic waters (Table 4, Figure 7C).

This same pattern was also observed in particulate backscattering, which is consistent with the presence of more phytoplankton particles in the water column that can backscatter light, as well as the presence of suspended matter coming from land to the ocean (Figure 7D).

**Table 4.** Median inherent optical properties measured across different reef zones compared to values derived offshore from MODIS satellite imagery for the similar time period.

| Reef Type | $a_g(440)$ (m$^{-1}$) | $a_{nap}(440)$ (m$^{-1}$) | $a_{ph}(440)$ (m$^{-1}$) | $b_{bp}(532)$ (m$^{-1}$) |
|---|---|---|---|---|
| Offshore (MODIS) | $0.004 \pm < 0.001$ [a] | | $0.009 \pm 0.001$ | $0.002 \pm < 0.001$ |
| Fore Reef | $0.038 \pm 0.015$ | $0.013 \pm 0.007$ | $0.022 \pm 0.007$ | $0.006 \pm 0.002$ |
| Back Reef & Lagoon | $0.046 \pm 0.021$ | $0.015 \pm 0.009$ | $0.030 \pm 0.007$ | $0.001 \pm 0.003$ |
| Fringing Reef & Terrestrial | $0.076 \pm 0.028$ | $0.082 \pm 0.065$ | $0.044 \pm 0.029$ | $0.021 \pm 0.014$ |

[a] The satellite-derived value is the summation of $a_g$ and $a_{nap}$, as these parameters have similar spectral shapes.

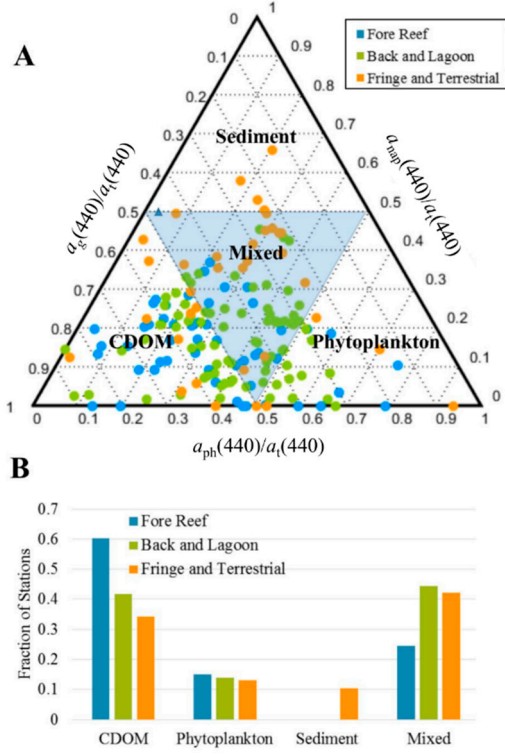

**Figure 8.** (**A**) Ternary plot showing the fraction of total non-water absorption dominated by colored dissolved organic matter (CDOM) ($a_g$), phytoplankton ($a_{ph}$), sediment ($a_{nap}$), and mixed (center triangle) for stations categorized according to the three geomorphic reef zones. (**B**) The fraction of stations from each geomorphic zone where total non-water absorption is dominated by one of the three constituents shown in panel A or mixed between the constituents (center triangle).

If we evaluate the relative contribution of absorption from phytoplankton, CDOM, and non-algal particles (sediments) at each station and zone, the majority of stations were influenced primarily by CDOM (Figure 8A).

This is particularly evident in the more oceanic waters of the fore reef stations where 60% of the stations were influenced by CDOM. The back/lagoon and fringing/terrestrial zones were more of an equal mixture between CDOM-dominated and waters influenced by equal parts of CDOM and phytoplankton (Figure 8B). The fringing/terrestrial zone was the only one where sediment occasionally was the dominant absorber consistent with the presence of fluvial influences. Benthic sediments on coral reef banks tend to be white colored with high scattering and low absorption properties [47,88].

The IOP values for each of the reef conditions can be related to biogeochemical parameters such as concentrations of Chlorophyll *a* (Chl) and Total Suspended Matter (TSM) using the bio-optical algorithms from Equations (5) and (7). For all stations, the median Chl is 0.13 mg m$^{-3}$ and the median TSM is 0.56 g m$^{-3}$, indicating fairly low particles on average, but with values generally increasing from oceanic to terrestrially influenced sites (Figure S12).

### 3.3. Optical Shape Parameters Across Geomorphic Zones

In addition to the magnitudes of the IOPs, the spectral shapes are useful in describing the composition of materials in the water (Figure 9).

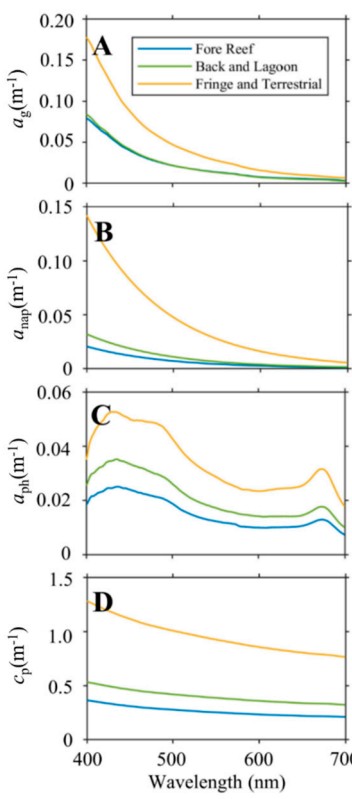

**Figure 9.** Mean spectra across geomorphic reef zones for absorption by CDOM (**A**), sediments (**B**), phytoplankton (**C**), and particulate attenuation (**D**).

Absorption by CDOM (Figure 9A), non-algal particles or sediments (Figure 9B), phytoplankton (Figure 9C) and particulate attenuation (Figure 9D) increased from oceanic to terrestrial reef sites. The median spectral slope of CDOM, $S_g$, determined with an exponential fit, was 0.0135 nm$^{-1}$. The slope of CDOM absorption ($S_g$) at fore reefs (median 0.0127 nm$^{-1}$) was similar to that of fringing and terrestrial reefs (median 0.0126 nm$^{-1}$) while back reef and lagoon locations were higher (median 0.0147 nm$^{-1}$). The median slope of particulate attenuation ($\gamma$) across all sites was 0.666 and also

increased significantly across geomorphic types, from fore reefs (median 0.621), to lagoon sites (0.664) and terrestrially influenced sites (0.687).

The shape of the volume scattering function in the backward direction (90°–180°) is of considerable interest to remote sensing studies [89–91] Here, the phase function was measured at three angles in the backward hemisphere (104°, 130° and 151°) and compared to other measurements and published relationships (Figure 10).

While only three angles are available for comparison, the values are similar to the mean results found across diverse oceanic regimes shown by Sullivan and Twardowski [91] and within analytical solutions following Mie theory and Fournier-Forand [89]. The phase function is fairly flat between 130° and 151°. While not inconsistent with the shape of the Petzold [92] phase function for clear, coastal or turbid waters, the data is qualitatively better fit to Sullivan and Twardowski [91].

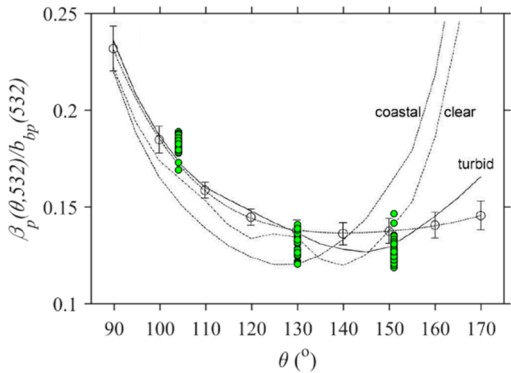

**Figure 10.** Angular particulate scattering measured at three angles in the backward direction ($\theta = 104°$, 130°, and 151°) normalized to the backscattering coefficient at 532 nm ($\beta(\theta,532)/b_{bp}(532)$) compared to published measurements and models of the phase function in the backward direction (90°–180°). The solid line with error bars depicts measurements from Sullivan and Twardowski [91] across diverse water types and the coastal, clear and turbid phase function are from Petzold [92]. Figure adapted from [91].

## 4. Discussion

This study represents the first integrative examination of coral reef water optical properties across geomorphic zone. We compare a broad range of reefs across the Central and Western Pacific Ocean, covering multiple geomorphic classifications. Light penetration and quality are determined by absorption and scattering in the water column. Broadly speaking, absorption determines the color of the water, i.e., which wavelengths are transmitted, while scattering is related to turbidity. While the Inherent Optical Properties define these parameters, a discussion of reef optics in terms of dissolved and particulate fractions may be more useful in understanding the dynamics governing light transmission and availability.

### 4.1. Colored Dissolved Organic Matter

The absorption characteristics of Pacific coral reef waters are dominated by absorption due to CDOM, $a_g$. The magnitude of $a_g(440)$ was similar across two of the three broad geomorphic zones, with the exception of terrestrially influenced sites (higher) (Figure 7). Similarly, CDOM was the dominant component of absorption at fore reef sites, and slightly under half of the lagoon and terrestrially influenced sites, with the majority of these two reef zones being mostly mixed-type. This is consistent with the findings of several studies [14,93]. Many models of benthic light availability in coastal systems are based largely on absorption by chlorophyll [94], which is clearly insufficient for CDOM-driven systems such as coral reefs.

Dissolved organic matter is strongly absorbing in the blue and ultraviolet (UV) wavelengths (Figure 3). Exposure to UV radiation can lead to genetic damage in corals, and CDOM plays an

important photoprotective role [2,10–12,95–97]. The depth of penetration for UV will be strongly regulated by both CDOM and suspended absorbing particulates, with potentially damaging irradiance levels observed in the first 10 m of the water column [98–100]. On Malaysian reefs, Kuwahara et al. [13] reported values of $a_g(440)$ between 0.09 and 0.59, with a mean of approximately 0.25 m$^{-1}$. This is considerably higher than the values observed here, as the mean of all stations was 0.06 m$^{-1}$, and the median was 0.05 m$^{-1}$. The values reported here are consistent with those measured over coral reefs of the Bahamas Banks [25]. Our results are also consistent with a study by Blondeau-Patissier et al. [93] in Australian waters that reported similar $a_g(440)$ values (0.014–0.083 m$^{-1}$).

Playing an important role in aquatic photochemistry and heating rates of the surface ocean, CDOM is found in all natural waters and results from the breakdown products of plant and other organic matter into humic materials [101,102]. The slope of CDOM absorption can provide information on the composition, source, photobleaching or microbial degradation state of CDOM [103–107]. In the absence of photobleaching, the spectral slope of CDOM can provide information on the source of materials where "humic" substances tend to have lower slopes (0.01) and "fulvic" substances have a steeper slope (0.02). For example, a low $S_g(\approx0.012$ m$^{-1})$ was found in Australian reefs that was characteristic of "humic" composition, which is suggestive of allochthonous production from the reef itself [93]. Here, the values of $S_g$ were lower on fore and fringing reefs and closer to the in situ reef production value of Blondeau-Patissier [93], with $S_g$ for lagoon locations being higher and similar to the "typical" marine values of 0.0145–0.015 nm$^{-1}$ [66], which could be correlated to photobleaching. Zanardi-Lamardo et al. [108] similarly found decreasing $S_g$ between coastal and reef stations. The state of CDOM on a given reef location will be determined by the interaction of considerable in situ generation, photobleaching, microbial degradation, and advection on and off the reef.

## 4.2. Particulates

Despite the importance of absorption by CDOM to coral reef optics, $a_{ph}$ or $a_{nap}$ may well dominate for some reefs or seasons depending on local conditions [9,109], which was also observed in the present study. The back reef and lagoon regions had more stations with absorption that was mixed between CDOM and phytoplankton than the fore reef stations (Figure 8B). Water column phytoplankton will absorb Photosynthetically Active Radiation (PAR), limiting light available to benthic primary producers. In addition to scattering and absorbing light, high levels of non-algal particles can be detrimental to corals, effectively clogging feeding systems or smothering them [51,110–112]. Personal observations confirmed that while some terrestrially influenced sites had considerable healthy coral cover, most had a bottom consisting of silt or organic sediments. It should be noted that while fringing reefs are attached to land and thereby influenced, the water column conditions at a given location may be more similar to lagoon conditions than other "terrestrial" zone sites. Here, the fringing/terrestrial zone had generally higher backscattering coefficients and stations where absorption was dominated by non-algal particles, likely sediment.

Water column particulate load is an extremely important factor in coral ecology [48,50,111,113,114]. Particulate matter can also be selectively removed from the water column by benthic filter feeders such as corals and sponges (e.g., [115]). The values of both algal and non-algal particulates observed during this study are within those described for other reef systems [116–118]. The median value of estimated total suspended matter across all sites (0.56 g m$^{-3}$) appears to be within a "normal" range for scleractinian coral reefs and is below that considered stressful [118–120].

Reef particulates are likely dominated by hard, inorganic (carbonate) particles. First, the measured ratio of backscattering to total scattering $b_{bp}/b_p$ provides information on particulate composition [20], with higher values indicating inorganic particles, and a lower ratio associated with organic, low-refractive-index particles like phytoplankton (Loisel et al. 2007) or marine snow/detritus [121,122]. The values of $b_{bp}/b_p$ (532 nm) found in this study (averaging ~0.03) are relatively high for some coastal areas [22]. However, Dierssen et al. [47] found waters over reef systems of the Bahama Banks dominated by fine carbonate sediments had similar average backscattering ratios.

The highest median values of $b_{bp}/b_p$(532 nm) in the data set were associated with back and fore reefs, which also had lower values of $a_{ph}$(440 nm). This may be related to wave action, which in turn, would lead to resuspension of particulates such as carbonate sands.

The estimate of the bulk index of refraction, derived from the backscattering ratio, provides information on the composition of the particulate population, with phytoplankton showing low values ($\approx$1.02–1.05) relative to seawater, while mineral particulates have much higher ($\approx$1.15–1.24) values [74,122]. At both fore and lagoon sites, the median derived $\hat{n}_p$ was 1.23 (Figure S9), which would be consistent with high levels of calcite (n = 1.24) in the bulk particulates [122]. Terrestrially influenced sites by contrast had the lowest ($p < 0.05$) $n_p$ values with a median of 1.18, more indicative of terrestrially sourced particulates such as clays (n = 1.18) and feldspar (n = 1.16).

The backscattering spectral slope $Y$ is expected to approach zero for a population of large, polydispersed particles [74,123], such as those in estuarine waters [22], while smaller particles and monodispersion would lead to higher values of $Y$ [22,124]. The $Y$ values here are fairly high (median 0.957), which suggests the predominance of smaller particles. Using the relationships developed by Slade and Boss [70], the high values of $Y$ suggest average particle diameters <10 μm. Fore reefs (median 1.011) and lagoon sites (median 0.995) showed higher values of $Y$, and therefore smaller, more uniform particulates than those in fringing and terrestrial reefs (median 0.771) (Figure S11).

### 4.3. Impact on Light Availability

These results suggest that the spectral quality and magnitude of light is quite different on a reef system compared to in the open ocean. The high levels of CDOM compared to offshore waters indicate that ultraviolet and blue light are absorbed rapidly with depth and more green light would reach the benthos. This could influence the composition of pigments within the corals and the amount of carotenoids that absorb more of the green light [125,126]. However, the presence of CDOM could also play a photoprotective role for the coral. High levels of irradiance can lead to oxidative damage of the dinoflagellate photosystems and bleaching of the coral, particularly in combination with elevated temperature [127,128]. The potential for such bleaching is increased when elevated temperatures prevent enzymatic repair, and corals in high-illumination conditions could become unable to tolerate them under projected ocean warming [128]. The release of CDOM could serve to protect the corals from exposure to high levels of ultraviolet light.

The spectral quality of light on the reef could also impact other photosynthetic primary producers in addition to corals, such as turf, calcareous, and fleshy algae [129–132], benthic diatoms [133], and seagrasses [134]. These organisms are likewise dependent on the availability of Photosynthetically Active Radiation (PAR) penetrating the water column and would need to similarly acclimate to the lower blue light penetration due to higher levels of CDOM.

### 5. Conclusions

Here, we provide a snapshot of the optical properties of coral reefs measured across diverse regions in the Pacific to provide insight into the light environment found in these dynamic ecosystems. We show that a range of IOPs exist across Pacific coral reef waters, from nearly oceanic to heavily attenuating coastal regimes. Overall, these data provide further evidence that coral reef systems themselves influence the water quality and spectral composition of light on the reef ecosystem. Reef waters from all sampled regions had substantially higher amounts of absorption and backscattering compared to the optical properties estimated offshore. Hence, simple bio-optical parameterizations cannot be used to describe the reef light environment and more nuanced models are required that incorporate variable contributions of CDOM, phytoplankton and sediments.

The variability in IOPs was influenced by the reef geomorphic zonation, as the physical processes that impact ecology ultimately shape development of the reef structure. Significant differences were observed across the three broad groupings of geomorphic zonation transitioning from more open ocean to more coastal in nature. Colored Dissolved Organic Material (CDOM) was the dominant component

of absorption across the majority of sites. Differences were observed based on the reef zonation with the clearer waters generally being in the fore reefs, and the back/lagoon regions were subject to more phytoplankton. Fringing reefs and terrestrial-impacted reefs exhibited the largest range in IOPs where the light environment could be dominated by the presence of CDOM, phytoplankton and sediments.

Coral reef biogeochemistry is largely driven by benthic primary production, and therefore, the intensity and spectral quality of light reaching the bottom. While external factors such as terrestrial sediment and nutrient delivery are important in determining the overall light regime, here, we have shown that coral reefs themselves may substantially alter local optical properties through the potential release of colored organic matter, even in the clearest tropical waters. This result was also demonstrated over the Bahamas Banks; benthic organisms like coral and seagrass influence the water column CDOM depending on the distance from the seafloor, tidal cycle, and mixing [25]. However, due to the confluence of high light and photobleaching in these shallow surface waters, determining the source of CDOM from the spectral slope measurements was challenging. Local production by both water column and benthic primary producers (i.e., "autochthonous") is likely to be a major source of CDOM. However, CDOM can also be released from sediment pore water during sediment resuspension events [68] and a substantial fraction can also be advected from nearby terrestrial sources ("allochthonous"), particularly in the fringing reef and terrestrially impacted zones. More research is needed to quantify the composition of CDOM across the various reef zones and evaluate the fate of CDOM in relation to physical and biogeochemical processes.

Significant advances have been made in remote sensing techniques to monitor the benthic composition from satellite and airborne sensors [8,53,57,135]. In particular, semi-analytical techniques have been developed to assess the optical properties of the water column along with the bathymetry and benthic composition from the water-leaving reflectance signal directly (Garcia et al. 2015; Hedley et al. 2016, Garcia et al. 2018). These techniques require high-quality hyperspectral reflectance measurements that have been accurately corrected for atmospheric absorption and scattering, as well as reflections of diffuse and direct light from the sea surface. The accuracy of these methods to retrieve water column optical properties have only been demonstrated across limited sites [136,137] and further research is needed to evaluate the uncertainties in retrievals of absorption and particulate backscattering from reefs and other optically shallow environments.

The use of multi-spectral satellite imagery with high spatial resolution has also proved useful for coral reef applications [135,138]. Such algorithms generally require simplifications such as *a priori* estimates of the water column optical properties [136]. The tendency is to use optically deep water as a proxy for shallow-water coral reef systems (e.g., [58]). Here, we quantitatively demonstrate that this assumption is not an accurate approach across most shallow-water reef ecosystems. Optically deep waters have substantially lower absorption and backscattering coefficients compared to water overlying shallow coral reefs. If no ancillary optical data is collected, then using the median optical properties derived here based on the geomorphic type of reef would be more accurate than using optically deep water as a proxy for the CDOM-enriched waters of the reef.

Despite decades of in situ and laboratory-based surveys and experimental study, coral reefs are not fully understood as ecosystems across spatial scales [3,139–142]. In addition to satellites and aircraft, using optical techniques from a diversity of platforms including divers [25,143,144], autonomous underwater vehicles [145], ships and kayaks [55], and drones [146] will also help to bridge this gap and unlock new methods to quantify physiology, metabolism, primary production, calcification and other biogeochemical processes. The complex three-dimensional nature of the benthos also influences light absorption [147] and reflectance [148]. The interplay between reef organisms and the surrounding aquatic environment is indeed complex, with many feedbacks that influence the light environment. This research advances the knowledge of the optical properties of the water column over shallow reefs of the Pacific with implications for assessing light and biogeochemistry across different reef zones.

**Supplementary Materials:** The following are available online at http://www.mdpi.com/2072-4292/11/15/1757/s1. Figure S1: Optical measurement stations during Kaneohe Bay 2016 campaign, Hawaii, USA; Figure S2: Optical

measurement stations at Lizard Island, Queensland, Australia during September 2016; Figure S3: Optical measurement stations at Heron Island, Queensland, Australia during September 2016; Figure S4: Optical measurement stations in Kaneohe Bay, Hawaii, USA during February 2017; Figure S5: Optical measurement stations in Guam, USA during April 2017; Figure S6: Optical measurement stations in Palau during May 2017; Figure S7: Volume scattering function normalized to back scattering; Figure S8: Slope of absorption by CDOM calculated using hyperbolic model ($S_h$) across campaigns and geomorphic classes; Figure S9: Mean Inherent Optical Properties across reef classifications; Figure S10: Water column absorption parameters across geomorphic classes; Figure S11: Water column scattering parameters across geomorphic classes; Figure S12: Derived bulk index of refraction, total suspended matter, and chlorophyll concentration across broad reef zones; Table S13: Minimum, maximum, and median value of optical parameters across geomorphic zones collected during this study, File S14: Mean collected data by class.

**Author Contributions:** Conceptualization, B.J.R., H.M.D., and E.J.H.; Data curation, B.J.R. and H.M.D.; Formal analysis, B.J.R.; Funding acquisition, E.J.H.; Investigation, B.J.R.; Methodology, B.J.R., and H.M.D.; Project administration, H.M.D., E.J.H.; Resources, E.J.H.; Software, B.J.R.; Supervision, H.M.D.; Writing—original draft, B.J.R. and H.M.D.; Writing—review & editing, B.J.R., H.M.D., and E.J.H.

**Funding:** Funding was provided by the COral Reef Airborne Laboratory (CORAL) project of the National Aeronautical and Space Administration and the Ocean Biology and Biogeochemistry program (NASA, grant number NNX16AB05G).

**Acknowledgments:** The authors wish to thank support staff and collaborators at Hawaii Institute of Marine Biology, University of Queensland, Coral Reef Research Foundation (Palau), Lizard Island Research Station, and Heron Island Research Station, as well as other members of the CORAL validation team. We thank Malcolm McFarland (Harbor Branch Oceanographic Institute) for absorption partitioning code. We acknowledge the NASA Ocean Biology Processing Group for access to processed MODIS imagery. We thank two anonymous reviewers whose comments were invaluable in improving this manuscript.

**Conflicts of Interest:** The authors declare no conflict of interest. The funders had no role in the design of the study; in the collection, analyses, or interpretation of data; in the writing of the manuscript, or in the decision to publish the results.

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
