# Peer review of "Water Column Optical Properties of Pacific Coral Reefs Across Geomorphic Zones and in Comparison to Offshore Waters"

_remotesensing, doi:10.3390/rs11151757_

Round 1
Reviewer 1 Report
This is a well written paper that is based on an extensive and detailed body of work by the authors. The paper explores in complete and thorough detail the spectral distribution of absorption and scattering in the waters above coral reefs and their breakdown in their component sources. The implications of their results are extremely significant in that they show that the common assumption used in remote sensing over reefs that the optical absorption and scattering spectra and their amplitudes are the same as for the open waters away from the reefs is invalid. Their studies also produce typical mean spectra that would be more reliable and considerably improve the analysis of reef health status from airborne or satellite sources. Their result are also given in terms of the various reef zones which makes them a useful reference for the development of any dynamic models of the nonlinear interaction between the geophysical and biological environment and the illumination of the reef which is key to estimating the survival probabilities of the coral. For the reasons above I recommend publication of this paper
There are some points which I have noted below which however need to be addressed before publication and would in my opinion improve the overall usefulness of the paper to the community.
1) line 156: apg and cpg are not defined and the symbols and their definitions should be included in Table I
2) line 190: at a minimum a brief outline of the method for separating anap and aph from ap must be given. This is a very important part of the results and as it stands the reader is forced to consult both references and is still not sure of what parts of the references are relevant and were actually used in the paper.
3)line 224: the reference for the definition of hyperbolic slope should be (Eq.4) and not (Eq.6)
4) line 395: from figure 10 the author’s results are not inconsistent with Petzold clear and turbid waters as they state. It is however a fair statement that they seem better fitted by the Sullivan 2009 results. Perhaps some numerical estimate of the goodness of fit could be given to confirm
Author Response
Reviewer comments in black, responses in blue.
This is a well written paper that is based on an extensive and detailed body of work by the authors. The paper explores in complete and thorough detail the spectral distribution of absorption and scattering in the waters above coral reefs and their breakdown in their component sources. The implications of their results are extremely significant in that they show that the common assumption used in remote sensing over reefs that the optical absorption and scattering spectra and their amplitudes are the same as for the open waters away from the reefs is invalid. Their studies also produce typical mean spectra that would be more reliable and considerably improve the analysis of reef health status from airborne or satellite sources. Their result are also given in terms of the various reef zones which makes them a useful reference for the development of any dynamic models of the nonlinear interaction between the geophysical and biological environment and the illumination of the reef which is key to estimating the survival probabilities of the coral. For the reasons above I recommend publication of this paper
Thank you, we certainly hope this will be useful to the community.
There are some points which I have noted below which however need to be addressed before publication and would in my opinion improve the overall usefulness of the paper to the community.
1) line 156: apg and cpg are not defined and the symbols and their definitions should be included in Table I
This has been corrected.
2) line 190: at a minimum a brief outline of the method for separating anap and aph from ap must be given. This is a very important part of the results and as it stands the reader is forced to consult both references and is still not sure of what parts of the references are relevant and were actually used in the paper.
Thank you for pointing this out. We have added a brief description of the methodology for clarification which explains the principle behind the methodology.
3)line 224: the reference for the definition of hyperbolic slope should be (Eq.4) and not (Eq.6)
This has been corrected.
4) line 395: from figure 10 the author’s results are not inconsistent with Petzold clear and turbid waters as they state. It is however a fair statement that they seem better fitted by the Sullivan 2009 results. Perhaps some numerical estimate of the goodness of fit could be given to confirm
Considering the low angular resolution of the data, we are unaware of an appropriate goodness of fit metric. We have added a line acknowledging the point about Petzold, with a potentially better fit to Sullivan 2009.
Reviewer 2 Report
Comments on the text:
Line 61/62: technically corals are animals and so not primary producers, you refer to corals later as being filter feeders (correct) so maybe clarify this as the coral zooxanthellae;
Line 69: You define PAR as Photosynthetically Available Radiation and later as Photosynthetically Active Radiation – it should be the latter so please edit for consistency;
Line 77: weird spacing, please check
Line 174: Please review this sentence, it doesn’t read that well so please edit for clarity;
Line 309: Missing full stop after ‘sites’
Line 302: Check the -1 subscript
Line 310: This is ambiguous, was it done twice, once in summer and once in winter, or twice in summer and twice in winter – you mean the former so a comma will help clarify this.
Table 4: Check the alignment of the columns so that the data sits under each other
Line 443: Closing quotes missing
Line 446: Definition of PAR again
Line 465 and 468 / 469 – add the units nm to the wave lengths
Line 550: (e.g., [58]) should be (e.g. [58]) – no comma
Equations: These seem to have been done in another package and added as images and as a result they are not great, for example Equation 2 is clipped. All of these can be done in word either as direct text or using the equation editor so suggest that this is done to improve the readability of these.
Figures: Same comment, these can be done in Excel or in a graphing package and imported as objects not as an image, some of the figures are very hard to see, for example 5b the colours are so similar that you can’t tell two of the three classes apart, Fig 8a ditto, Fig 8b the same and Figure 10 is also hard to read. I suggest re-doing these in excel or Illustrator (or Inkscape) and importing them as objects so that they print, you also need to choose colours that are distinct in a mono print out.
Comment on the Paper:
I liked the paper and the central tenant that deep water signals do not apply to shallow systems such as reefs. I really, really, really, really wish you had taken some off reef or ocean samples at the same time as the on reef ones rather than using MODIS so it was a like for like comparison. I almost gave it the thumbs down because of this as it is frustrating that you couldn’t directly compare the on reef and ocean samples at the same time using the same method as that would have been much more robust. I have worked at Heron and Lizard and it is possible to only go a short distance and be away from the reef so I feel this was a major omission.
I also had some trouble with:
· I didn’t see where you label which sites were in each of the geomorphological zones – it is hard to see from Fig 5 as the colours are the same. For example, are the sites at Lizard fringing reef or not?? The time of sampling means that you won’t get any run-off from the island so they will act as oceanic reefs but technically they are fringing. Ditto for Hawaii and even more so with two seasons being measured so the reefs may act as terrestrial in some seasons but not in others.
· Figure 10 is very hard to read as the quality is low, I think you need to re-do this or import the graph as objects not as an image.
Author Response
Reviewer comments in black, responses in blue.
Comments on the text:
Line 61/62: technically corals are animals and so not primary producers, you refer to corals later as being filter feeders (correct) so maybe clarify this as the coral zooxanthellae;
You bring up a good point, which gets to the common vernacular in the field. This has been clarified in the text.
Line 69: You define PAR as Photosynthetically Available Radiation and later as Photosynthetically Active Radiation – it should be the latter so please edit for consistency;
This has been corrected.
Line 77: weird spacing, please check
This has been corrected.
Line 174: Please review this sentence, it doesn’t read that well so please edit for clarity;
This line has been clarified.
Line 309: Missing full stop after ‘sites’
This has been corrected.
Line 302: Check the -1 subscript
This has been corrected.
Line 310: This is ambiguous, was it done twice, once in summer and once in winter, or twice in summer and twice in winter – you mean the former so a comma will help clarify this.
This line has been clarified.
Table 4: Check the alignment of the columns so that the data sits under each other
Spacing has been corrected, however, if this refers to the “Offshore (MODIS)” row, the data in the first column is intentionally placed between the ag and anap columns because the satellite data is the summation of these two. There is a note in the table pointing this out.
Line 443: Closing quotes missing
This has been corrected.
Line 446: Definition of PAR again
This has been corrected.
Line 465 and 468 / 469 – add the units nm to the wave lengths
This has been corrected.
Line 550: (e.g., [58]) should be (e.g. [58]) – no comma
This has been corrected here and at another line.
Equations: These seem to have been done in another package and added as images and as a result they are not great, for example Equation 2 is clipped. All of these can be done in word either as direct text or using the equation editor so suggest that this is done to improve the readability of these.
Thank you for catching this. In fact, these were all done in the word equation package, but several appear to have been altered when pasted into the journal template. This has been corrected.
Figures: Same comment, these can be done in Excel or in a graphing package and imported as objects not as an image, some of the figures are very hard to see, for example 5b the colours are so similar that you can’t tell two of the three classes apart, Fig 8a ditto, Fig 8b the same and Figure 10 is also hard to read. I suggest re-doing these in excel or Illustrator (or Inkscape) and importing them as objects so that they print, you also need to choose colours that are distinct in a mono print out.
Again, this appears to be an import/paste issue has been corrected by changing the insertion method. Extra attention will be paid during final proof to ensure quality.
Comment on the Paper:
I liked the paper and the central tenant that deep water signals do not apply to shallow systems such as reefs. I really, really, really, really wish you had taken some off reef or ocean samples at the same time as the on reef ones rather than using MODIS so it was a like for like comparison. I almost gave it the thumbs down because of this as it is frustrating that you couldn’t directly compare the on reef and ocean samples at the same time using the same method as that would have been much more robust. I have worked at Heron and Lizard and it is possible to only go a short distance and be away from the reef so I feel this was a major omission.
We completely understand and share the frustration about the lack of measured off-shore data. Unfortunately, due to weather and boat limitations during field campaigns, we were unable to get far enough off-shore to reach water that was clearly “offshore;” after careful deliberation we decided not to create confusion or error by labeling samples that were likely still influenced by the reef as being “oceanic.”
I also had some trouble with:
· I didn’t see where you label which sites were in each of the geomorphological zones – it is hard to see from Fig 5 as the colours are the same. For example, are the sites at Lizard fringing reef or not?? The time of sampling means that you won’t get any run-off from the island so they will act as oceanic reefs but technically they are fringing. Ditto for Hawaii and even more so with two seasons being measured so the reefs may act as terrestrial in some seasons but not in others.
We have corrected our import method and the figures should be much clearer, but we will make any further changes as needed during final layout.
You bring up an excellent point about sampling time and tidal cycle influencing the “classification” of the reef. We decided to base our classifications more on morphology than water column effects (which will, as pointed out, change over relatively short time scales). In fact, this will be the subject of an upcoming manuscript.
· Figure 10 is very hard to read as the quality is low, I think you need to re-do this or import the graph as objects not as an image.
This has been corrected (as stated above) by changing our import method.